# Estimated Glomerular Filtration Rate as a Prognostic Factor in Urothelial Carcinoma of the Upper Urinary Tract: A Systematic Review and Meta-Analysis

**DOI:** 10.3390/jcm10184155

**Published:** 2021-09-15

**Authors:** Min Hyuk Kim, Hyeong Dong Yuk, Chang Wook Jeong, Cheol Kwak, Hyeon Hoe Kim, Ja Hyeon Ku

**Affiliations:** Department of Urology, Seoul National University College of Medicine, Seoul National University Hospital, Seoul 03080, Korea; michael920@hanmail.net (M.H.K.); hinayuk@naver.com (H.D.Y.); drboss@gmail.com (C.W.J.); mdrafael@snu.ac.kr (C.K.); hhkim@snu.ac.kr (H.H.K.)

**Keywords:** prognosis, upper urinary tract, urothelial carcinoma, renal function, renal insufficiency

## Abstract

Preoperative renal function is associated with worse outcomes in patients undergoing radical nephroureterectomy (RNU). The purpose of this systemic review and meta-analysis was to determine the association of preoperative renal function with oncological outcome in patients who underwent RNU. We searched articles published up to March 2021 in PubMed, Scopus, and Embase by combining “urothelial carcinoma”, “radical nephroureterectomy”, and “estimated glomerular filtration rate”. We also manually screened the reference list for publications following general guidelines recommended by the preferred reporting items for systematic reviews and meta-analyses (PRISMA) statement. The relationship between preoperative renal function and survival was expressed as overall survival (OS), progression-free survival (PFS), and cancer-specific survival (CSS). This review and meta-analysis included 13 studies involving a total of 4668 patients who received RNU. Pooled analysis showed significant negative association of preoperative renal function with PFS (HR: 1.51, 95% CI: 1.23–1.80, *p* < 0.00001), CSS (HR: 1.63, 95% CI: 1.38–1.92, *p* < 0.00001), and OS (HR: 1.22, 95% CI: 1.10–1.35, *p* < 0.00001). Patients with upper tract urothelial carcinoma (UTUC) who received RNU showed a significant negative association of preoperative renal function with survival.

## 1. Introduction

Tumors originating in urothelial cells, including ureter to renal pelvis, are known as upper tract urothelial carcinomas (UTUCs). UTUCs are rare malignant tumors that account for approximately 5–10% of all urothelial cancers [1,2,3]. Radical nephroureterectomy (RNU) with bladder cuffing represents the ultimate treatment for highly recurrent UTUC [4,5]. In previous studies, the classification of five-year cancer-specific survival (CSS) was based on pathologic stages. The 5-year CSS exceeded 90% when the final pathological result was pT1 (non-muscle invasive cancer) or less. However, the 5-year CSS decreased to 40% when the pathological result was pT3 or higher [6]. This finding suggests that RNU is sufficient for organ-confined early-stage UTUC, whereas RNU alone is insufficient for non-organ confined advanced UTUC with non-organ confined or lymph node metastasis. Complete lymph node dissection along with RNU can increase CSS in patients with UTUCs (pT3 or higher) [7,8,9,10]. Other studies reported that adjuvant chemotherapy with locally advanced UTUC (pT3N0/Nx, pT4N0/Nx, or pTanyN+) can effectively increase CSS [11]. Based on the findings of previous studies, locally advanced UTUCs are an indication for lymph node dissection and adjuvant chemotherapy.

Several prognostic factors for UTUC have been reported. Postoperative pathological parameters such as pathologic tumor stage (pT), lympho-vascular invasion (LVI), tumor grade, tumor necrosis, lymph node (LN) involvement, surgical margin, and histological variants are strong prognostic factors. Preoperative prognostic factors include the presence of hydronephrosis, serum CRP, tumor size, tumor location, history of previous bladder cancer, age, Eastern Cooperative Oncology Group performance status (ECOG PS), and chronic kidney disease (CKD) [12,13,14,15,16,17,18,19,20,21]. However, most of the studies reporting various prognostic factors were retrospective in design.

CKD is a common disease diagnosed in the elderly population. It is associated with malignancies of kidney and ureter [22,23]. The underlying treatment for UTUC is radical resection of the kidney, leaving the patient with a unilateral kidney for survival after surgery. As the unilateral kidney after the surgery needs to perform the function of both kidneys, the patient’s renal function might be reduced compared with the level before the surgery, which could result in CKD and affect survival [24,25] Renal function is particularly important in patients with locally advanced UTUC because of the need for adjuvant chemotherapy after surgery. Although many studies have shown that CKD is a significant preoperative prognostic factor in UTUC, most of these studies are limited by their retrospective format. Therefore, the objective of this study was to analyze the association of preoperative renal function with postoperative survival using a systematic review and meta-analysis.

## 2. Materials and Methods

This systematic review and meta-analysis was conducted and reported in accordance with the general guidelines recommended by the preferred reporting items for systematic reviews and meta-analyses (PRISMA) statement [26]. We reviewed studies involving participants who underwent radical nephrectomy for UTUC, including open laparoscopic and robotic surgical interventions.

### 2.1. Data Sources and Search Strategy 

The PRISMA flow chart is shown in Figure 1. We searched PubMed, Scopus, and Embase for related articles published before March 2021. The following search terms and their combinations were used: “urothelial carcinoma”, “radical nephroureterectomy”, and “estimated glomerular filtration rate”. We also manually screened the reference lists of publications to identify potentially relevant studies for analysis.

### 2.2. Inclusion and Exclusion Criteria

We used strict inclusion and exclusion criteria to limit the heterogeneity of the entire study. If the study met the following including criteria, it was eligible for additional evaluation: (1) patients diagnosed with urothelial carcinoma via pathological testing; (2) determination of relationship between preoperative renal function and prognosis; and (3) studies describing the hazard ratio (HR) and 95% confidence interval (CI) using multivariate survival analysis.

The exclusion criteria were: (1) letters, comments, case reports, reviews, and conference abstracts with limited data; (2) publications in languages other than English; (3) studies performed using animals or cell lines; and (4) duplicate articles and articles with duplicate data. If the same patient population was evaluated in several studies, only the latest or the largest study was included in the analysis. Studies that did not report adjusted HR using multivariate analysis were excluded because of uncertain accuracy of HR without multivariate analysis. For studies that leveraged both multivariate and univariate analyses to estimate clinical outcomes, results of the multivariate analysis were used to calculate HR and CI. Each study was independently screened by two reviewers (M.H.K. and J.H.K.) to determine compliance with the selection criteria. Any disagreement was resolved by consensus. Finally, 13 papers [7,11,22,27,28,29,30,31,32,33,34,35,36] were included in this study.

### 2.3. Data Extraction

Two investigators (H.D.Y. and C.W.J.) reviewed each eligible article individually and extracted information from all publications that met the inclusion criteria. Information was retrieved based on the Reporting Recommendations for Tumor Marker Prognostic Studies (REMARK) involving prognostic markers. The data table was configured to extract all relevant data contained in each study text, table, or figure. Disagreement was resolved through discussion.

### 2.4. Quality Assessments

Methodological quality of each study was evaluated independently by three reviewers (M.H.K., C.K. and H.H.K.) using the Newcastle–Ottawa scale (NOS) including cohort studies only (Appendix A). A maximum score of 1 was assigned to each item and a maximum score of 2 was allowed only for comparability. Therefore, the final quality score varied from 0 (the lowest) to 9 (the highest), and a total score of 0–5 was considered low, 6–7 was considered intermediate, and 8–9 was considered high quality.

### 2.5. Definition of Survival

**Definition** **1.***Survival is defined as follows. Cancer-specific survival (CSS) is the period of survival until death from cancer after surgery, and progression-free survival (PFS) is the period of recurrence of cancer at the surgical site, metastasis to nearby lymph nodes, and metastasis to distant sites. Overall survival (OS) represents the survival period after surgery*.

### 2.6. Statistical Analysis

Survival data were compiled based on the time-to-event occurrence HR for operating measurements. The HRs and 95% CIs were calculated using a random-effects model. Forest plots were used to estimate the effect of eGFR on patient survival and disease progression. A statistical test of heterogeneity was performed based on the Q-test and I2 test to assess heterogeneity during the study [37]. A *p*-value > 0.05 and an I2 < 50% were considered non-heterogeneous. Potential publication bias was assessed through visual inspection of the funnel plot. Statistical significance was defined at the 0.05 level. All statistical analyses were performed using RevMan 5.4.1 software (Cochrane Collaboration, Cochranehagen, London, UK).

## 3. Results

Individual characteristics of 13 studies are presented in Table 1. The recruitment period was from 1991 to 2017. The number of patients included in these studies ranged between 70 and 666, with a total of 4668 patients. All 13 were retrospective studies. Of them, 11, 2, and 11 studies mentioned inclusion/exclusion criteria, definition of survival, and definition of eGFR, respectively.

Table 2 shows patient characteristics reported in each study. Median age ranged from 67 to 74 years. Three studies did not indicate age. Nine studies described a surgical approach. An open approach was used for 2031 cases and a laparoscopic approach was adopted for 2393 cases. The median follow-up period ranged from 16 to 65 months.

Tumor and pathologic characteristics are listed in Table 3 and Table 4, respectively. Ten studies reported tumor location. The renal pelvis harbored 1830 tumors and the ureter carried 1783 tumors. In nine studies, a total of 656 people received adjuvant chemotherapy. In 11 studies, tumor grade was low in 1335 (31%) cases and high in 2950 (69%) cases. The pathologic T stage was described in nine studies, including 2381 (60%) below pathologic T2 and 1577 (40%) higher than T3.

Table 5 presents results of multivariate analyses using the Cox proportional hazards model. Survival analysis was expressed as overall survival (OS), progression-free survival (PFS), or cancer-specific survival (CSS). The standard of eGFR was set at 30 mL/min/1.73 m^2^ in one study, 50 mL/min/1.73 m^2^ in two studies, and 60 mL/min/1.73 m^2^ in the remaining studies.

Figure 2 presents a forest plot and a funnel plot to demonstrate PFS according to preoperative renal function. Eight studies were included. Five studies showed significant positive association between PFS and preoperative renal function (adjusted HR: 1.51; 95% CI: 1.23–1.80, *p* < 0.00001). The funnel plot was relatively symmetrical, showing no evidence of significant publication bias.

Nine studies showed a relationship between CSS and preoperative renal function. The forest plot and funnel plot are shown in Figure 3. All but one study showed a positive association between CSS and preoperative renal function. Six studies showed a significant positive association between CSS and preoperative renal function (adjusted HR: 1.63, 95% CI: 1.38–1.92, *p* < 0.00001). The funnel plot was relatively symmetrical, without strong evidence for publication bias.

Seven studies showed a relationship between OS and preoperative renal function. Forest plots and funnel plots are shown in Figure 4. All but one study showed a positive association between OS and preoperative renal function. Three studies showed a significant positive association between OS and preoperative renal function (adjusted HR: 1.22, 95% CI: 1.10–1.35, *p* < 0.00001). We could not find a strong evidence for publication bias based on the funnel plot.

## 4. Discussion

This study investigated whether renal function of patients with UTUC before RNU was associated with survival rate. Thirteen studies were included. The total number of patients was 4668. All patients were diagnosed with UTUC and, therefore, underwent RNU. Each study expressed the survival rate as CSS, PFS, or OS. The final meta-analysis showed that preoperative renal function was related to postoperative survival rate.

Ito et al. [7] analyzed 70 patients with N0M0 UTUC who underwent unilateral RNU between 1999 and 2012. The survival rate was expressed as a 3-year extraurothelial recurrence-free survival rate (EURFS). In the multivariate Cox proportional hazards model, the EURFS had a worse outcome in patients with a preoperative eGFR less than 60 mL/min/1.73 m^2^ (HR: 6.579, 95% CI: 1.934–22.222, *p* = 0.0026). Yeh et al. [35] investigated the postoperative prognosis according to the presence of preoperative hydronephrosis and flank pain in 472 UTUC patients who underwent RNU in a single medical center from 1991 to 2013. The survival rate was expressed as 5-year CSS and 5-year OS using the Kaplan–Meier method. The eGFR was set at 60 mL/min/1.73 m^2^. Those with preoperative hydronephrosis and flank pain had worse outcomes of 5-year CSS and 5-year OS, respectively. Since preoperative hydronephrosis and flank pain were associated with preoperative renal function, patients with eGFR less than 60 mL/min/1.73 m^2^ before surgery had worse outcomes of 5-year CSS (HR: 1.691, 95% CI: 1.071–2.669, *p* = 0.024) and 5-year OS (HR: 1.577, 95% CI: 1.045–2.382, *p* = 0.030). Ehdaie et al. [38] developed a model to predict the prognosis of 253 patients who underwent RNU for UTUC between 1995 and 2008. A multivariable Cox regression model was used and eGFR was set as a continuous variable. Survival rates were expressed as 5-year CSS and 5-year PFS. The higher the preoperative eGFR, the better was the 5-year PFS (HR: 0.73, 95% CI: 0.61–0.88, *p* < 0.001) and 5-year CSS (HR: 0.74, 95% CI: 0.61–0.90, *p* = 0.002). 

These preceding studies showed that preoperative renal function was positively correlated with CSS, PFS, and OS of patients with UTUC. However, Xylinas et al. [28] showed no association between preoperative renal function and survival rate of patients with UTUC who underwent RNU. Xylinas et al. investigated 781 patients with UTUC treated with RNU from 1994 to 2007 at seven different centers. The preoperative eGFR criterion was set at 60 mL/min/1.73 m^2^, and the postoperative eGFR criterion was set at 45 mL/min/1.73 m^2^. Univariable and multivariable Cox regression models were used. Neither preoperative nor postoperative eGFR was associated with 5-year CSS, PFS, or OS. In our systematic review and meta-analysis, eight studies demonstrated a relationship between PFS and preoperative eGFR (adjusted HR: 1.51, 95% CI: 1.23–1.80, *p* < 0.00001). In five studies, PFS and preoperative eGFR showed a significant positive correlation. Although one study showed a positive correlation and two studies showed a negative correlation, all three studies showed no significant correlations. Nine studies showed a relationship between CSS and preoperative eGFR. Although one study showed a negative correlation between CSS and preoperative eGFR and two studies showed a positive correlation, none of them showed statistically significant correlation between CSS and preoperative eGFR. The remaining six studies showed a significant positive relationship between the two (adjusted HR: 1.63, 95% CI: 1.38–1.92, *p* < 0.00001). Seven studies showed a relationship between OS and preoperative eGFR. Although one study showed a negative relationship, the correlation was not significant. Six studies showed a positive relationship between OS and preoperative eGFR. However, only three studies showed significant correlation between the two variables (adjusted HR: 1.22, 95% CI, 1.10–1.35, *p* < 0.00001). Results of this study confirmed that the preoperative renal function of patients was closely related to their survival rate after RNU.

Several previous studies have shown that renal function decreases after kidney surgery [28,39,40]. Although patients who underwent radical nephrectomy had severe renal impairment more than those who underwent partial nephrectomy, the rate of CKD was increased postoperatively in patients with partial nephrectomy [41]. In addition, UTUC patients who underwent radical nephrectomy had significantly higher serum creatinine increase and higher rates of ESRD hemodialysis than those of RCC patients (HR: 2.9, 95% CI: 1.88–4.49, *p* < 0.001) [39]. Some studies have shown that patients with CKD or ESRD exhibited a lower survival rate than that of those with normal renal function [24,25]. If UTUC patients manifest reduced preoperative renal function, they carry a high probability of developing CKD or ESRD due to their decreased renal function after radical nephrectomy-based RNU. Therefore, it can be inferred that they will have poor outcomes such as disease prognosis and survival rate.

Patients with non-organ-confined or lymph node metastasis undergoing UTUC require adjuvant chemotherapy because it is impossible to perform surgical treatment. Adjuvant chemotherapy for UTUC basically entails a combination of gemcitabine and cisplatin [42,43]. Cisplatin-induced nephrotoxicity is well known [44,45]. When cisplatin is absorbed into renal tubular cells, it can trigger an inflammatory response via multiple signaling pathways, leading to histological damage. Cisplatin also affects renal vessels and causes ischemic damage [46]. Therefore, patients with reduced renal function cannot use cisplatin-based chemotherapy and, thus, exhibit a worse survival rate.

Our systematic review and meta-analysis study has several limitations. First, 10 out of 13 studies involved Asians including five from Japan, three from Taiwan, and one each from Korea and China [7,22,27,29,30,32,33,34,35,36]. The remaining three papers included two from a multinational study and one from the United States [11,28,31]. The number of patients included 3343 Asians, 245 Americans, and 1080 patients from multiple nations. Thus, Asians accounted for at least 71% of all subjects. Since Asians were included in the study performed in the United States and different nations, it limited the evaluation of other ethnic groups. Second, all 13 papers included in this study were retrospective studies. However, most of these studies had an adequate sample size and a NOS quality score of 6 or higher. Finally, there was heterogeneity between papers. The I2 value indicating heterogeneity exceeded 50% (PFS, 64%; CSS, 56%). Thus, the results of this systematic review and meta-analysis require careful interpretation.

In conclusion, this systematic review and meta-analysis revealed that patients with decreased eGFR before surgery manifested poor PFS, CSS, and OS after RNU. However, all studies included in this meta-analysis were retrospective in nature. Thus, a large-scale prospective study is needed in the future.

## Figures and Tables

**Figure 1 jcm-10-04155-f001:**
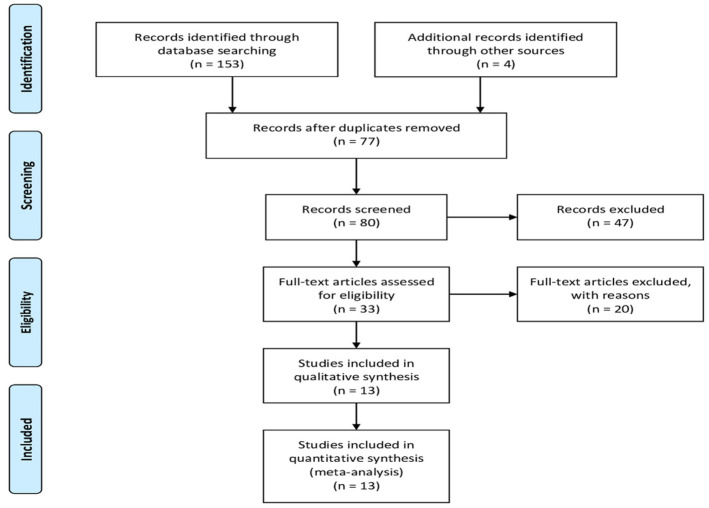
A PRISMA flowchart of the literature search strategy used in our meta-analysis and systemic review.

**Figure 2 jcm-10-04155-f002:**
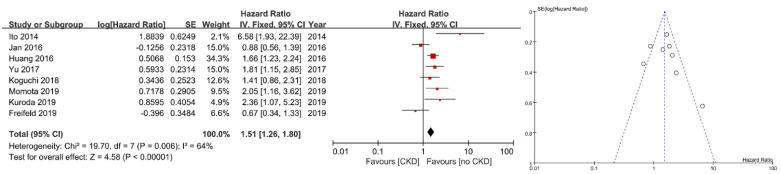
Forest plot and funnel plot of progression-free survival after radical nephroureterectomy according to preope-ative renal function. SE: standard error; IV: inverse variance; CI: confidence interval; df: degree of freedom.

**Figure 3 jcm-10-04155-f003:**
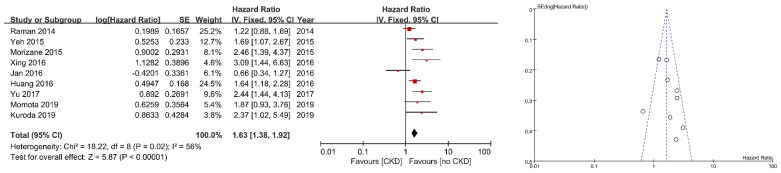
Forest plot and funnel plot of cancer-specific survival after radical nephroureterectomy according to preparative renal function.

**Figure 4 jcm-10-04155-f004:**
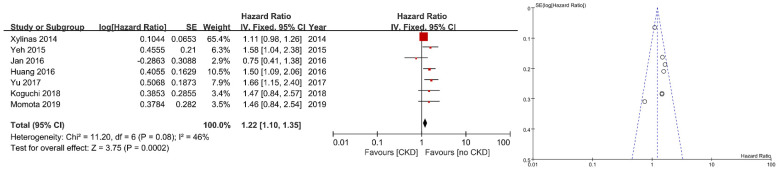
Forest plot and funnel plot of overall survival after radical nephroureterectomy according to preoperative renal function.

**Table 1 jcm-10-04155-t001:** Main characteristics of the eligible studies.

Study	Year	Country	Recruitment Period	No. of Patients	Prospective Data Collection	Inclusion/Exclusion Criteria	Consecutive Patients	Definition of Survival	Definition of eGFR
Xylinas	2013	Multination	1994–2007	666	No	Yes	NA	Yes	Yes
Ito	2014	Japan	1999–2012	70	No	Yes	NA	No	Yes
Raman	2014	Multination	2003–2012	414	No	No	NA	No	Yes
Morizane	2015	Japan	2000–2012	345	No	Yes	NA	Yes	No
Yeh	2015	Taiwan	1991–2013	472	No	Yes	NA	No	Yes
Huang	2016	Taiwan	2001–2016	198	No	Yes	NA	No	Yes
Xing	2016	China	2000–2013	192	No	Yes	NA	No	Yes
Yu	2017	Korea	2004–2014	566	No	Yes	NA	No	Yes
Koguchi	2018	Japan	1990–2015	433	No	Yes	NA	No	Yes
Freifeld	2019	USA	1993–2016	245	No	Yes	NA	No	No
Jan	2019	Taiwan	2007–2017	424	No	Yes	NA	No	Yes
Kuroda	2019	Japan	1999–2017	187	No	No	NA	No	Yes
Momota	2019	Japan	1995–2017	456	No	Yes	NA	No	Yes

eGFR: estimated glomerular filtration rate; NA: not available.

**Table 2 jcm-10-04155-t002:** Patient characteristics of the eligible studies.

Study	Median Age, Range (Years)	Gender (Male/Female)	Median BMI, Range (kg/m^2^)	ECOG Performance Status (0/1/2/3)	Smoking	Surgical Approach (Open/Laparoscopic)	Median Follow-Up, Range (Months)
Xylinas	69.6, 54–76	441/225	28.2, 24–32 (IQR)	445/221 (1–3)	NA	519/147	45.5, 24–67 (IQR)
Ito	NA	47/23	NA	NA	NA	49/21	29.2, 1–157
Raman	70, 27–96	257/157	NA	82/165/159/8	NA	NA	16, 2–120
Morizane	74, 38–95	234/111	22.1, 13–34.2	241/103 (1–3)	175	244/101	39.9, 6.1–160
Yeh	67, 24–95	204/268	NA	NA	99	269/203	33, 1–233
Huang	68.6, 23.6–91.6	103/95	NA	NA	26	NA	29.1, 6.4–164.9
Xing	NA	78/114	NA	NA	NA	84/145	65, 3–144
Yu	72, 65–76 (IQR)	165/401	NA	399/148/20/1	NA	142/424	31.1, 16.2–55.7
Koguchi	69, 62–75 (IQR)	313/120	NA	NA	138	243/190	35.4, 13.8–74.5 (IQR)
Freifeld	70 (mean)	152/93	29 (mean)	126/98 (1–3)	NA	NA	27
Jan	70, 29–96	189/235	NA	NA	49	NA	35, 14–60 (IQR)
Kuroda	71, 38–90	138/49	NA	NA	NA	104/83	49.2, 3.4–209.2
Momota	NA	309/147	NA	446 (0–1)/10 (2–3)	NA	377/79	40

BMI: body mass index; ECOG: Eastern Cooperative Oncology Group; IQR: interquartile range; NA: not available.

**Table 3 jcm-10-04155-t003:** Tumor characteristics of the eligible studies.

Study	History of Bladder Cancer	Hydronephrosis	Tumor Size	Tumor Location (Pelvis/Ureter)	Tumor Multifocality	Adjuvant Chemotherapy
Xylinas	244	NA	NA	420/246	164	62
Ito	17	26	NA	0/70	7	23
Raman	NA	NA	NA	NA	NA	55
Morizane	36	201	NA	140/205	51	NA
Yeh	NA	NA	NA	189/193	90	87
Huang	49	NA	NA	NA	NA	21
Xing	NA	119	NA	102/90	52	NA
Yu	111	249	NA	258/308	49	205
Koguchi	NA	NA	NA	239/194	NA	99
Freifeld	80	71	3.4 (mean)	116/85	35	NA
Jan	127	344	NA	191/138	116	40
Kuroda	47	112	NA	NA	NA	NA
Momota	NA	288	NA	175/254	NA	64

NA: not available.

**Table 4 jcm-10-04155-t004:** Pathologic characteristics of the eligible studies.

Study	Tumor Grade (Low/High)	Pathologic T Stage (pT0/Is/a/1/2/3/4)	Pathologic N Stage (pNx/−/+)	Variant Form	LVI	Concomitant CIS	Positive Surgical Margin
Xylinas	121/533	326 (≤T1)/118/182/40	291/291/84	NA	171	229	NA
Ito	NA	NA	NA	NA	NA	NA	NA
Raman	116/298	3/16/106/60/60/143/26	165/203/46	NA	NA	NA	25
Morizane	222/109	188 (≤T2)/152 (≥T3)	205/119/21	29	102	43	22
Yeh	112/360	0/60(Tis/a)/130/112/142/28	261/170/41	8	NA	NA	NA
Huang	11/147	198 (T3)	198 (N0)	NA	31	20	5
Xing	170/22	30 (Ta)/162 (T1)	192 (N0)	NA	NA	NA	NA
Yu	182/388	0/84 (Tis/a)/128/134/200/20	NA	NA	119	54	NA
Koguchi	300/127	0/18/66/91/81/153/24	181/221/31	NA	151	NA	30
Freifeld	NA	NA	NA	NA	NA	NA	NA
Jan	22/402	0/0/161 (Ta/1)/83/180 (T3/4)	399 (Nx/−)/25	NA	115	NA	NA
Kuroda	55/132	96 (≤T2)/91 (≥T3)	0/172/15	NA	65	22	21
Momota	24/432	260 (≤T2)/196 (≥T3)	431 (Nx/−)/25	NA	160	NA	15

LVI: lymphovascular invasion; CIS: carcinoma in situ; NA: not available.

**Table 5 jcm-10-04155-t005:** Estimation of hazard ratios.

Study	Survival Analysis	Threshold of eGFR (mL/min/1.73 m^2^)	Co-Factors	Analysis Results
Xylinas	OS	60	Standard clinico-pathological features	Not significant
Ito	PFS	60	cT stage (T3), length of cancer (3 cm), maximal diameter of cancer (1.6 cm), NLR (3)	Significant
Raman	CSS	60	Gender, race, age (70 years), ECOG performance status (0,1/2,3), pT stage (T3/T4), LN status, surgical margin status, adjuvant chemotherapy	Not significant
Morizane	CSS	50	ECOG performance status (0/≥1), number of tumor (1/>1), CRP (0.5)	Significant
Yeh	CSS/OS	60	Gender, age (67 years), smoking, surgery method (laparoscopic/open), tumor location, pT stage, pN stage, tumor grade, adjuvant chemotherapy, hematuria, hydronephrosis and flank pain	Significant/Significant
Huang	PFS/CSS/OS	60	Gender, age (68.6 years), current smoking, ASA score, recurrent bladder tumor, recurrent contralateral UTUC, tumor grade, LVI, CIS, surgical margin status, adjuvant radiotherapy, adjuvant chemotherapy	Significant/Significant/Significant
Xing	CSS	30	ABCC6 methylation, GDF15 methylation, tumor multifocality, surgery method (laparoscopic/open)	Significant
Yu	PFS	60	BMI, pT stage (≤T2/≥T3), tumor grade, LVI	Significant
	CSS	60	DM, pT stage (≤T2/≥T3), tumor grade, LVI, adjuvant chemotherapy	Significant
	OS	60	Age, BMI, ECOG performance status(0,1/2,3), tumor size, tumor multifocality, pT stage (≤T2/≥T3), tumor grade, LVI, adjuvant chemotherapy	Significant
Koguchi	PFS/OS	60	Change rate of eGFR, age, gender, tumor location, tumor grade, pT stage, pN stage, LVI, surgical margin status	Not significant/Not significant
Freifeld	PFS	50	Age (66 years), ECOG performance status (0/≥1), hemoglobin, hydronephrosis, pT stage (≤T2/≥T3), tumor architecture	Not significant
Jan	PFS/CSS/OS	60	Gender, blood type, age (69 years), smoking, hemodialysis, DM or hypertension, previous or concomitant bladder cancer, hydronephrosis, hematuria, pT stage, pN stage, tumor grade, LVI, tumor location, tumor multifocality, tumor size (3 cm), tumor architecture, tumor necrosis, adjuvant chemotherapy, NLR (4), PLR (150), MLR (0.4), SII (580)	Not significant/Not significant/Not significant
Kuroda	PFS	60	Tumor histology, pT stage (≤T2/≥T3), tumor grade, pN stage, surgical margin status, LVI, CAR (0.079, NLR (2.035), PLR (165), GPS (1), fibrinogen (337)	Significant
	CSS	60	Urine cytology, tumor histology, pT stage (≤T2/≥T3), tumor grade, pN stage, surgical margin status, LVI, CAR (0.079, NLR (2.035), PLR (165), GPS (1), fibrinogen (337)	Significant
Momota	PFS/CSS/OS	60	age, gender, ECOG performance status, hypertension, CVD, DM, neoadjuvant chemotherapy, hydronephrosis, tumour location, tumor grade, pT stage (≤T2/≥T3), pN stage, LVI	Significant/Not significant/Not significant

eGFR: estimated glomerular filtration rate; OS: overall survival; PFS: progression-free survival; NLR: neutrophil-to-lymphocyte ratio; CSS: cancer-specific survival; ECOG: Eastern Cooperative Oncology Group; LN: lymph node; CRP: C-reactive protein; ASA: American Society of Anesthesiologists; UTUC: upper tract urothelial carcinoma; LVI: lymphovascular invasion; CIS: carcinoma in situ; BMI: body mass index; DM: diabetes mellitus; PLR: platelet-to-lymphocyte ratio; MLR: monocyte-to-lymphocyte ratio; SII: systemic immune-inflammation index; CAR: C-reactive protein-to-albumin ratio; GPS: Glasgow prognostic score; CVD: cardiovascular disease.

## Data Availability

The data presented in this study are available on request from the corresponding author.

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
