# Peer review of "Estimated Glomerular Filtration Rate as a Prognostic Factor in Urothelial Carcinoma of the Upper Urinary Tract: A Systematic Review and Meta-Analysis"

_jcm, 2021, doi:10.3390/jcm10184155_

Round 1

Reviewer 1 Report

Authors wanted to determine the association of preoperative renal function with the oncological outcome of patients who underwent RNU through a systematic review and meta-analysis, by taking into consideration overall survival (OS), progression free survival (PFS) and cancer specific survival (CSS). The research question has a relevance because patients with locally advanced disease need adjuvant chemotherapy and even because CKD that may result as patient live on unilateral kidney affect survival.

I have few concerns.

  • In the section inclusion and exclusion criteria, authors assess that “... sufficient information was included to estimate the hazard ratio ....” (Lines 86-87); they should better clarify what they mean for sufficient information.
  • Results: I think that when you consider a study for a meta-analysis related to cancer survival and eGFR as prognostic factor, you need to have a mention of both definitions, so if just two studies show this kind of definitions this could be not sufficient.
    As mentioned by the authors, eGFR may be affected by adjuvant chemotherapy and thus by pT stage: why did you considered in your meta-analysis studies where the pT stage was not considered, or where it was not specified if patients were submitted to post-operative chemotherapy? This may lead to a major bias; you should consider a homogeneous population (I.e.: patients who underwent neo-adjuvant chemotherapy/ patients who underwent adjuvant chemotherapy/ patients who did not chemotherapy and standardize them for clinical and pathological stage.
  • Line 255: add to citation [].
  • I agree with the authors that this study has several limitations relatively to the Asiatic prevalence of the sample, to the fact that it is a retrospective study, and to the heterogeneity.

Author Response

Response to Reviewer 1 Comments

Point 1: In the section inclusion and exclusion criteria, authors assess that “... sufficient information was included to estimate the hazard ratio ....” (Lines 86-87); they should better clarify what they mean for sufficient information.

 Response 1: We agree with the reviewer’s comment. In response to the reviewer’s comment, we added the sentence, as follows. This sentence is written in the sense that we included papers using multivariate analysis rather than univariate analysis. Reflecting the editor's opinion, we will revise it as follows: “(3) studies describing the hazard ratio (HR) and 95% confidence interval (CI) using multivariate survival analysis”.

Point 2: I think that when you consider a study for a meta-analysis related to cancer survival and eGFR as prognostic factor, you need to have a mention of both definitions, so if just two studies show this kind of definitions this could be not sufficient.

Response 2: We agree with the reviewer’s comment. In response to the reviewer’s comment, we added the sentence, as follows. As shown in Table 1, of the 13 studies included in our study, the definition of eGFR was mentioned in 11 studies, but the definition of survival was mentioned in only 2 studies. Based on previous studies, the definition of survival is defined as follows. Cancer-specific survival (CSS) is the period of survival until death from the cancer after surgery, and progression-free survival (PFS) is the period of recurrence of the cancer at the surgical site, metastasis to nearby lymph nodes, and metastasis to distant sites. Overall-survival (OS) represents the survival period after surgery. We will add the above information in the manuscript.

Point 3: As mentioned by the authors, eGFR may be affected by adjuvant chemotherapy and thus by pT stage: why did you considered in your meta-analysis studies where the pT stage was not considered, or where it was not specified if patients were submitted to post-operative chemotherapy? This may lead to a major bias; you should consider a homogeneous population (I.e.: patients who underwent neo-adjuvant chemotherapy/ patients who underwent adjuvant chemotherapy/ patients who did not chemotherapy and standardize them for clinical and pathological stage.

Response 3: We agree with the reviewer’s comment. Our study compared survival with preoperative eGFR. Adjuvant chemotherapy is administered after surgery, so it is not related to preoperative eGFR. Neo-adjuvant chemotherapy can affect renal function before surgery, so we agree with the editor's comment. However, in our study, only one paper out of 13 papers performed neo-adjuvant chemotherapy, only PFS had a meaningful result, and CSS and OS had meaningless results. We do not think it will have a significant impact on the meta-analysis results.

Point 4: Line 255: add to citation [].

Response 4: We agree with the reviewer’s comment. In response to the reviewer’s comment, we added [] to the citation of line 255.

Point 5: I agree with the authors that this study has several limitations relatively to the Asiatic prevalence of the sample, to the fact that it is a retrospective study, and to the heterogeneity.

Response 5: We agree with the reviewer’s comment. In response to the reviewer’s comment, we also considered the above limitations. If more North American and European data are reported and prospective studies are conducted in the future, it is necessary to proceed with an updated meta-analysis.

Reviewer 2 Report

The authors present an interesting meta-analysis making the correlation between renal function before surgery and survival in upper tract urothelial carcinoma such as prognostic factor in this setting. 

In general, the paper is well written and result are consistent with analysis realized.

Manuscript needs minor English review

Figure 2 and 3 should be remake because the letter is not easily readable. 

Line 170 and 180 yellow remark should be removed

The major concern in the paper is that the vast majority of studies analyzed were Asians. The authors asses it correctly in the study limitations

Author Response

Response to Reviewer 2 Comments

Point 1: Manuscript needs minor English review

 Response 1: We agree with the reviewer’s comment. In response to the reviewer’s comment, we will perform English proofreading and submit the manuscript.

Point 2: Figure 2 and 3 should be remake because the letter is not easily readable.

Response 2: We agree with the reviewer’s comment. In response to the reviewer’s comment, we resubmit with higher quality of Figures 2 and 3.

Point 3: Line 170 and 180 yellow remark should be removed

Response 3: We agree with the reviewer’s comment. In response to the reviewer’s comment, we removed the yellow remark.

Point 4: The major concern in the paper is that the vast majority of studies analyzed were Asians. The authors asses it correctly in the study limitations

Response 4: We agree with the reviewer’s comment. In response to the reviewer’s comment, we also considered the above limitations, and if more North American and European data are reported in the future, it is necessary to proceed with an updated meta-analysis.